# A Case Report of a Feto-Placental Mosaicism Involving a Segmental Aneuploidy: A Challenge for Genome Wide Screening by Non-Invasive Prenatal Testing of Cell-Free DNA in Maternal Plasma

**DOI:** 10.3390/genes14030668

**Published:** 2023-03-07

**Authors:** Luigia De Falco, Giuseppina Vitiello, Giovanni Savarese, Teresa Suero, Raffaella Ruggiero, Pasquale Savarese, Monica Ianniello, Nadia Petrillo, Mariasole Bruno, Antonietta Legnante, Francesco Fioravanti Passaretti, Carmela Ardisia, Attilio Di Spiezio Sardo, Antonio Fico

**Affiliations:** 1AMES, Centro Polidiagnostico Strumentale, 80013 Naples, Italy; 2Fondazione Genetica per la Vita Onlus, Via Cuma, 80132 Naples, Italy; 3Department of Molecular Medicine and Medical Biotechnologies, Federico II University Hospital, Via Pansini 5, 80131 Naples, Italy; 4Department of Public Health, University of Naples “Federico II”, 80145 Naples, Italy; 5CRR Genetica Medica, Azienda Ospedaliera s. Maria della Misericordia, 06156 Perugia, Italy

**Keywords:** non-invasive prenatal testing, cfDNA, genome-wide screening, feto-placental mosaicism and single nucleotide polymorphisms (SNP) array

## Abstract

Non-invasive prenatal testing (NIPT) using cell-free DNA can detect fetal chromosomal anomalies with high clinical sensitivity and specificity. In approximately 0.1% of clinical cases, the NIPT result and a subsequent diagnostic karyotype are discordant. Here we report a case of a 32-year-old pregnant patient with a 44.1 Mb duplication on the short arm of chromosome 4 detected by NIPT at 12 weeks’ gestation. Amniocentesis was carried out at 18 weeks’ gestation, followed by conventional and molecular cytogenetic analysis on cells from the amniotic fluid. SNP array analysis found a *de novo* deletion of 1.2 Mb at chromosome 4, and this deletion was found to be near the critical region of the Wolf-Hirschhorn syndrome. A normal 46,XY karyotype was identified by G-banding analysis. The patient underwent an elective termination and molecular investigations on tissues from the fetus, and the placenta confirmed the presence of type VI true fetal mosaicism. It is important that a patient receives counselling following a high-risk call on NIPT, with appropriate diagnostic analysis advised before any decisions regarding the pregnancy are taken. This case highlights the importance of genetic counselling following a high-risk call on NIPT, especially in light of the increasing capabilities of NIPT detection of sub-chromosomal deletions and duplications.

## 1. Introduction

Cell-free DNA (cfDNA)-based prenatal screening, also known as non-invasive prenatal testing (NIPT), has high sensitivity and specificity for common autosomal trisomies [1,2]. The detection rates for trisomy 21 typically exceed 99% with low false-positive (FP) rates, <0.1%, [3] and rare false-negative (FN) cases reported in large clinical studies [4,5,6]. In recent years, the use of cfDNA to screen chromosome alterations has been expanded to genome-wide screening for rare autosomal aneuploidies (RAAs) and partial deletions and duplications (i.e., copy number variants, including selected microdeletions), and an increasing number of studies have described the test performance and the clinical validity of these additional fetal anomalies [7,8,9,10,11]. A recent study that looked at diagnostic outcomes for CNV cases identified by genome-wide NIPT found an overall PPV of >70% [12], while in another study a positive predictive value (PPV) of 22.4% for RAA was reported in a large cohort [9]. Although PPV for RAA is not as high as that for common trisomy, several studies demonstrated that several of the additional findings identified through genome-wide NIPT have a clinical impact [11].

However, NIPT cannot produce a diagnostic result, due to technical and computational limits and also to biological issues. Common biological causes of false-positive NIPT results include placental mosaicism [13], fetal chromosome rearrangements, a vanishing twin [14], and maternal chromosome abnormalities or malignancy [15]. An inherent limitation of NIPT, irrespective of the approach taken, is the discordance between NIPT and invasive prenatal diagnosis, mainly due to confined placental mosaicism (CPM). As such, cases with CPM are potential sources of false-positive (FP) and false-negative (FN) results, affecting the interpretation and management of NIPT results, since NIPT screening is used to determine the risk of fetal chromosome aneuploidies [16].

In about 2% of pregnancies, the cytogenetic constitution of the placenta may not match that of the fetus because of the biological phenomenon known as feto-placental mosaicism [17]. Two principal mechanisms, both involving a post-zygotic event, can generate a chromosomal mosaicism: non-disjunction in a somatic cell at mitosis, which produces two daughter cells, one trisomic with 47 chromosomes and the other one monosomic with 45 chromosomes; and trisomy rescue after a meiotic non-disjunction through the loss of one supernumerary chromosome in a somatic cell restoring a disomic cell line [18]. When the mitotic error occurs before the embryological separation between fetal and extra-fetal compartments, the chromosomal mosaicism will likely affect both placenta and fetus; if it occurs after this stage, it is likely to be confined to only one of them [19]. There are six main classes of mosaicism, including three classes of CPMs (Types I, II, and III) and three classes of TFMs (Types IV, V, and VI). These classes are based on whether the chromosome aberration is present in the cytotrophoblast only, the mesenchymal core only, or both placental tissues, and whether the aberration is seen in the placenta only or is also present in the fetus [17]. Several factors may impact the clinical effects of chromosomal mosaicism, including the size of the gene imbalance, the timing of the initial event, and the distribution of the abnormal cells in the tissue [20].

Very recently, Eggenhuizen et al. [21] performed a literature review looking at the association between CPM and adverse pregnancy outcomes. The authors found that CPM was associated with fetal growth restriction, preterm birth, structural fetal anomalies, and pregnancy complications such as preeclampsia. False positive results are confirmed by invasive prenatal testing: chorionic villous sample (CVS) and amniocentesis.The choice of CVS or amniocentesis depends on the chromosomal alteration identified by NIPT because of the possibility of false-positive or mosaic results when confirmatory testing is carried out using chorionic villi [2,22]. Little is known about the sensitivity of NIPT for the detection of CPM, especially because the studies on how cytogenetic results of NIPT relate to those from CVS and placenta are rare. Very recently, Van Opstal et al. reported four cases in which NIPT, CVS, and placenta cytogenetic data were combined and showed the higher sensitivity of NIPT for detection of CPM involving the cytotrophoblast as compared to CVS [23]. However, NIPT remains a screening test, which, as recommended by international medical genetics societies, should be accompanied by genetic counseling, especially in cases with high-risk cfDNA results, so that the results can be confirmed with an invasive diagnostic prenatal test [24,25], before making any pregnancy decisions. Here we present a case in which NIPT revealed a structural chromosome aberration and in which fetal, placental, and parental cytogenetic and cyto-genomic follow-up investigations during and after pregnancy, were performed in order to elucidate the discrepancy that was found between the abnormal NIPT (partial duplication of 4p chromosome) and differently abnormal fetal karyotype (4p deletion).

## 2. Case Report

The patient was a 32-year-old pregnant Caucasian woman referred to the AMES laboratory in Naples for NIPT. The first-trimester ultrasound findings were normal: nuchal translucency of 1.9 mm, crown rump length (CRL) of 62 mm, and fetal heart rate of 161 bpm. The pregnant woman reported apparent good health and was being treated for an autoimmune disease (Hashimoto’s thyroiditis, psoriasis, and autoimmune gastritis). Both the pregnant woman and her partner had a negative family history for genetic or chromosomal diseases. They did not report consanguinity. The couple had a first pregnancy with intrauterine growth restriction due to flow alterations and the patient went on to deliver a healthy baby (now 14 months old) at 40 weeks of gestation.

For NIPT analysis, a total of 10 mL peripheral blood was collected at 12 + 3 weeks and NIPT was carried out using VeriSeq^TM^ NIPT Solution v2 assay (Illumina Inc., San Diego, CA, USA) [8,26]. The pregnant woman chose the Genome-wide analysis report, used to receive results about common trisomies (13, 18 and 21) and sex chromosomes, as well as results on rare autosomal aneuploidies and partial deletions/duplications ≥ 7 Mb. NIPT analysis did not detect trisomy in chromosomes 21, 18, and 13. The presence of a Y chromosome was also reported. Moreover, NIPT results indicated a 44.1 Mb duplication on the short arm of chromosome 4: dup(4)(p16.3p12) (Figure 1 and Table 1), with a DNA fetal fraction at 11%.

Analysis of the supplementary NIPT report allowed us to obtain some values regarding this duplication, as reported previously [27,28] (Table 1). For example, the log likelihood ratio (LLR), which is calculated for each target chromosome, and each sample provides a determination of aneuploidy and should be reviewed with respect to the specified cut-off prior to interpreting the result with the VeriSeq NIPT Solution v2 assay [8]. In our case the “region_llr_trisomy” value was 148.38, which exceeded the cut-off value provided for CNVs (15.1).

Following post-test genetic counselling regarding the NIPT results, the patient agreed to undergo amniocentesis at 18 weeks of gestation and 15 mL of amniotic fluid was retrieved for cytogenetic and cyto-genomic analysis. SNP array analysis (HumanCytoSNP-12 BeadChip, Illumina), carried out on cells from the amniotic fluid as well as on cells from the peripheral blood of parents, found a *de novo* deletion of 1.2 Mb at chromosome 4: 4p16.3 (48,283-1,243,573) which included the genes *ZNF141* [OMIM*194648], PDE6B [OMIM*180072], *IDUA* [OMIM*252800], and *RNF212* [OMIM*612041] (Figure 2a). This deletion was found to be near the critical region of the Wolf-Hirschhorn syndrome (WHS). Considering the discordant results between NIPT and the SNP-array analyses, a possible feto-placental mosaicism was suspected and 40 metaphases from flask cultured amniocytes were analyzed with CytoVision software (CytoVision, AB Imaging). A normal 46,XY karyotype was identified by GTG-banding analysis (Figure 2b). Parental karyotypes, performed at that time, showed a normal karyotype. A fetal ultrasound was also carried out at the same time; however, no clear findings suggesting chromosomal abnormalities in the fetus were found.

The family was concerned about the results and, during the post-test counselling, they were informed about the limits of NIPT in detecting CNVs, and about the presence of a possible mosaicism which could explain the discrepancy between NIPT and the SNP array results. In addition, they were informed about the uncertain clinical significance of the de novo variant, in particular in predicting the clinical phenotype and the presence or the grade of the intellectual disability. The couple decided to terminate the pregnancy and agreed to perform molecular investigations on the tissues from the fetus and the placenta. The fetal dysmorphological examination showed a male fetus with dysmorphic facies characterized by the presence of hypertelorism, micrognathia and a cystic igroma in the neck. The autopsy confirmed normal intrathoracic, intraabdominal and pelvic organs. The placenta showed no macroscopic or histologic anomalies. Fetal skin DNA showed the same 1.2 Mb deletion (4p16.3 (48,283–1,243,573) found on cells from the amniotic fluid (Figure 3).

Regarding chromosomal testing of the placenta, SNP array analysis carried out on three placental samples showed different results. Samples P1 and P3 showed a 1.2 Mb deletion (arr (GRCh37) 4p16.3 (48,283–1,243,573) × 1) and a 26.5 Mb duplication, both on the chromosome 4p (arr (GRCh37) 4p16.3-p15.1 (1,285,521–27,805,588) × 3 mos) (Figure 4a,b). The estimated percent mosaicism of cells with the duplication was 60%. Sample P2 showed a 1.2 Mb deletion (arr (GRCh37) 4p16.3 (48,283–1,243,573) × 1) and a 4.6 Mb deletion, both on the short arm of chromosome 4 (arr (GRCh37) 4p16.3-p16.2 (1,260,337–5,907,731) × 1 mos) (Figure 4c). This latter deletion was found to be present in 70% of the tested cells. Considering these results, we conclude that this was a case of type VI true fetal mosaicism (TFM; anomaly in the cytotrophoblast, mesenchyme, and the fetus). A schematic chart of these two hypotheses is reported in Figure 5.

## 3. Discussion

Genome-wide NIPT using cell-free DNA can screen for a wide variety of chromosomal anomalies, including CNVs and RAA. Although this expanded NIPT screen is not currently recommended by professional medical societies, it is becoming increasingly available in clinics around the world. A recent study that looked at diagnostic outcomes for CNV cases identified by genome-wide NIPT found an overall PPV of >70% [12]. Another recent study from the Dutch NIPT Consortium [11] found that the majority of additional findings identified through genome-wide NIPT have a clinical impact. Most of the confirmed fetal aberrations in that study were found to be pathogenic, and the CPM (confined placental mosaicism)/assumed CPM cases were found to be significantly associated with adverse perinatal outcomes [11]. The study also found that an additional finding was reported in about one in every 275 pregnant patients that chose to undergo genome-wide NIPT, highlighting the importance of this type of screening.

However, as NIPT is a screening test, false-positive and false-negative results can occur, with feto-placental mosaicism likely to be the main biological cause for false-positive NIPT results [29]. Most frequently, a complete or partial aneuploidy and/or a chromosomal structural rearrangement may be present in one of the cell lines [30,31,32]. Chromosomal mosaicism is one of the primary interpretative issues in prenatal diagnosis and it is found through villocentesis and amniocentesis, in 1–2% and 0.1–0.3% of pregnancies, respectively [17,33,34]. Furthermore, the distribution of the different cell lines in the fetus and/or the placenta depends on the time when the mosaicism occurred and, on the embryo/fetal localization, as well as the type of chromosome mechanism. Mosaicism could involve (1) only the placenta, where the condition is known as “confined placental mosaicism”; (2) both the placental and the fetus, where the condition is “feto-placental mosaicism” [18,19]; (3) the fetus only.

As mosaicism is a main cause of false positives and false negatives with NIPT, an invasive diagnostic procedure should be carried out to confirm the presence or absence of the anomaly before any decisions regarding the pregnancy are taken. The choice of a CVS or amniocentesis for confirmation needs to be carefully considered, with factors such as time between NIPT and final diagnosis, the type of anomaly reported on NIPT, and the absence/presence of abnormal findings on an ultrasound scan, to be taken into account. As amniocentesis cannot be carried out until at least 15 weeks of gestation, this could result in prolonged anxiety for the patient, depending on when the initial NIPT was carried out. However, amniocentesis is considered the gold standard for confirmation of a fetal anomaly, as a CVS procedure analyzes placental tissues only, and cannot be used to rule out a fetal aberration [35]. If a patient does choose CVS over amniocentesis, it is important that both layers of the placenta are analyzed, as this would give the most accurate risk assessment for fetal involvement [35,36].

In our study, NIPT carried out at 12 + 3 weeks of gestation revealed a large duplication on the short arm of chromosome 4. In our case, SNP-array analyses highlighted a del(4)p16.3) in cultured amniocytes, in fetal skin and, as a mosaic, in placenta samples 1 and 3. In addition, in placenta samples 1 and 3, a dup(4)p16.3-p15.1) was observed. Placenta sample 2 showed both a del (4)p16.3 in 30% and a del (4)p16.2 in 70%.

The 4p16.3 deletion, observed in the fetus, probably could have not been detected with NIPT, given the resolution limit of the method and the size of the anomaly. Regarding the rearrangement-size threshold, it is established on 7 Mb by the “VeriSeq^TM^ NIPT Solution v2 assay”, so the 4p16.3 deletion of the fetus, of 1.2 Mb, was under this cutoff value and therefore was undetectable. The presence of both the 4p16.3 duplication and the 4p16.2 deletion in the placenta could be explained as placenta-confined mosaicism of segmental aneuploidies. In fact, CPM is defined as a chromosomally abnormal cell line restricted to the placenta, while the chromosomes of the fetus itself are normal [37]. CPM usually does not give rise to ultrasound abnormalities or pregnancy complications, so it is in general found by accident.

The latter evidence could suggest the hypothesis of an initial formation of a zygote with del 4p16.3 deletion, originating during the parental gametogenesis; later, in the stages of embryonic development, the other genomic rearrangements observed in samples 1, 2 and 3 of the placentas (del4p16.2 and dup4p16.3), due to non-allelic homologous recombination (NAHR), would have originated and been confined to the placenta (Figure 5A. An alternative hypothesis could be an initial formation of a zygote 46,XY, originating during normal parental meiotic events. Then, during the mitotic events of embryonic development, the del4p16.3, in the embryo, mos dup4p16.3-15.1/del4p16.3 (placenta P1 eP3) and mos del4p16.2/del4p16.3 (placenta P2) in placenta were observed, due to non-allelic homologous recombination (NAHR) (Figure 5B). Moreover, we cannot exclude that there may be other mechanisms, since we have not found similar cases in the literature.

The identification that ∼5% of the human genome consists of interspersed duplications with a high degree of identity at the nucleotide level (>95%) and covering large genomic distances has raised intense research interest in the dynamic mechanisms of mutation of the human genome and the role of these duplications in evolution [38]. Many segmental duplications, also called low-copy repeats, or duplicons, are present in our genome in every chromosome, with a nonuniform distribution [39,40]. The mechanisms that lead to the generation of segmental duplications are not completely understood, but it seems that they are preferentially located in sub-telomeric and pericentromeric regions [38]. Multiple DNA sequences throughout the genome are similar. In some cases, within the two sequences involved in a particular exchange, there is a length of perfect or near perfect homology, and this is the site of the actual strand exchange (NAHR). Inverted segmental duplications or highly identical low-copy repeat (LCR) sequences can mediate the formation of inversions and more complex structural rearrangements through NAHR. One of the main challenges in investigating mosaicism is to establish at which point during the embryo-fetal development the mitotic error occurs [18]. However, the majority of embryos at the cleavage stages consist of a mixture of cells with normal and abnormal chromosomal constitutions, or cells with different abnormalities [41].

These mosaic embryos are the result of post-zygotic errors, i.e., chromosome segregation errors occurring during the first mitotic divisions. In addition, the degree of mosaicism and the affected cell lineages, resulting in diverse mosaic patterns, depend on the timing of the segregation error [42]. In partial, terminal, or interstitial chromosomal deletions, the pathogenetic mechanism that determines the phenotype is haploinsufficiency. Partial chromosomal deletions are most often a de novo event: they occur mainly on the chromosome of paternal origin and correlate with advanced paternal age.

The *de novo* 4p16.3 deletion was found in a disease region near the critical region for Wolf-Hirshhorn syndrome but did not completely overlap and included a number of OMIM genes (*ZNF141*, *PIGG*, *IDUA*, *CPLX1*, *SLC26A1*, *RNF212*, *PDE6B*, and *CTBP1*). WHS is a contiguous gene deletion syndrome associated with a hemizygous deletion of chromosome 4p16.3 involving the two critical regions WHSCR1 and WHSCR2 [43] and is characterized by typical craniofacial features, prenatal and postnatal growth impairment, intellectual disability, severe delayed psychomotor development, seizures, and hypotonia [44]. WHS deletions greater than 3–5 Mb seem to be associated with a more severe phenotype and a higher risk of heart defects and cleft palate [45]. Hence, the prenatal diagnosis for WHS is difficult due to a large diversity of expression of this syndrome and rather non-specific ultrasound findings, such as intrauterine growth restriction or increased nuchal translucency [46]. In our case, as well as in cases where no clear ultrasound anomalies were found or could become evident only later during pregnancy, the expanded NIPT, increasingly used in microdeletion/duplication screening, is the only way to detect this type of chromosomal alteration earlier in the pregnancy.

In a review of the literature, eight case reports of derivative chromosome detected by NIPT and inherited from a previously undiagnosed parent have been reported [47]. The smaller CNV detected was a 7.9 Mb deletion and only a 708 Kb deletion was not detected by NIPT. Therefore, NIPT seems to be a good approach for the screening of CNVs greater than 7 Mb inherited from balanced translocation.

Here, follow-up testing of fetal and placental tissues following an elective termination confirmed the presence of type VI (or True fetal) mosaicism.

## 4. Conclusions

In summary, as the cfDNA in the maternal plasma fraction originates from the cytotrophoblast of chorionic villi, the presence of a feto-placental mosaicism has implications for the interpretation and management of cfDNA testing results for the risk assessment of fetal chromosome aneuploidies. It is important that patients with a screen-positive result for a chromosomal anomaly following NIPT are counselled appropriately and advised to undergo an invasive diagnostic procedure for confirmation before any decisions regarding the pregnancy are taken.

## Figures and Tables

**Figure 1 genes-14-00668-f001:**
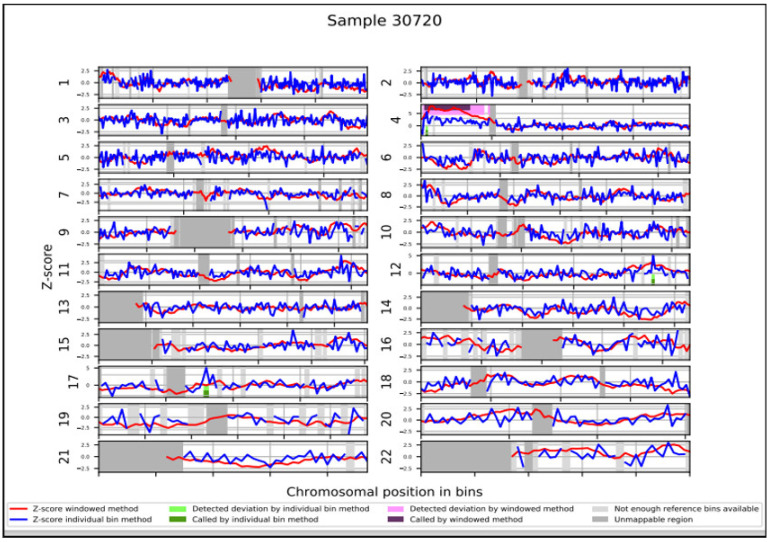
NIPT results. WISECONDOR plot showing the abnormal NIPT result with a duplication of short arm of chromosome 4.

**Figure 2 genes-14-00668-f002:**
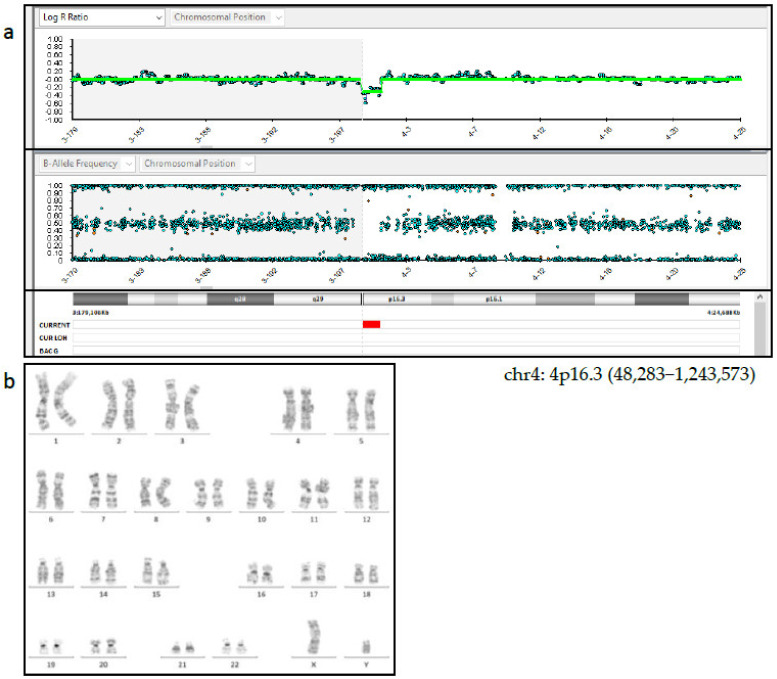
(**a**) SNP-array analysis of amniotic fluid showed a *de novo* deletion of 1.2 Mb at chromosome 4: 4p16.3 (48,283–1,243,573). (**b**) GTG banding on cultured cells from the amniotic fluid showed a 46,XY karyotype.

**Figure 3 genes-14-00668-f003:**
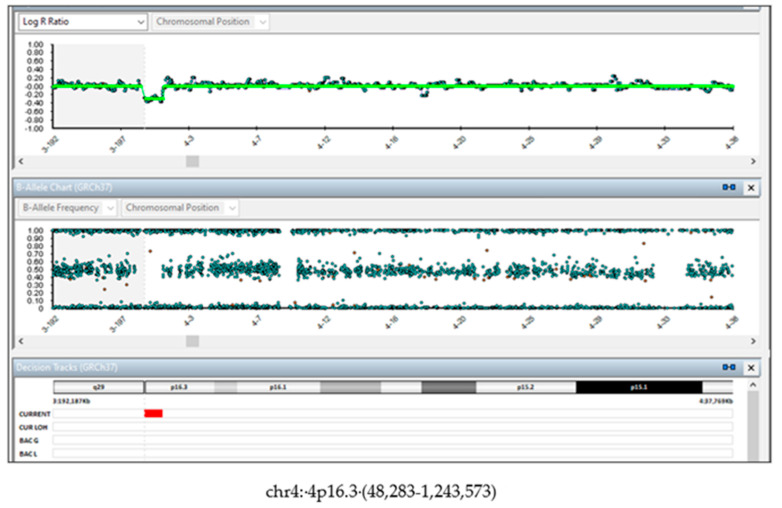
SNP-array on fetus tissue. DNA from fetal skin showed the *de novo* 1.2 Mb deletion (4p16.3 (48,283-1,243,573).

**Figure 4 genes-14-00668-f004:**
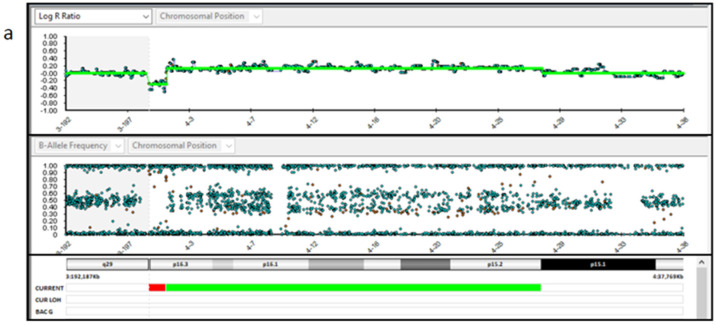
SNP-array analysis of the placenta tissues. (**a**,**b**). Placenta sample 1 and 3 DNA (P1 and P3) showed a 1.2 Mb deletion (arr ((GRCh37) 4p16.3−p15.1 (1,285,521–27,805,588) × 3 mos). (**c**). Placenta sample 2 DNA (P2) showed a 1.2 Mb deletion (arr (GRCh37) 4p16.3 (48,283–1,243,573) × 1) and a 4.6 Mb deletion, both on the chromosome 4p (arr (GRCh37) 4p16.3-p16.2 (1,260,337–5,907,731) × 1 mos).

**Figure 5 genes-14-00668-f005:**
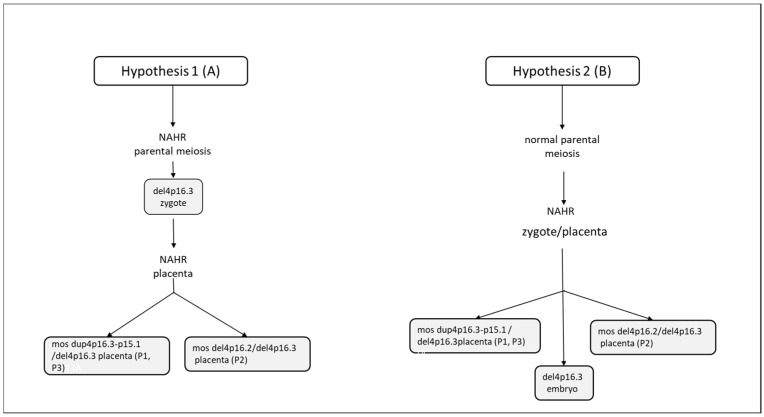
Schematic chart to explain the mosaic formation. (**A**) Initial formation of a zygote with del 4p16.3 deletion, originating during the parental gametogenesis; later, in the stages of embryonic development, the other genomic rearrangements observed in samples 1, 2 and 3 of the placenta (dup(4p16.3−p15.1 and del(4p16.2), due to non-allelic homologous recombination (NAHR) and, subsequent to a mitotic non-disjunction, would have originated in and been confined to the placenta. (**B**) Initial formation of a zygote 46,XY, originating during parental meiotic events. Then, during the mitotic events of embryonic development, the del4p16.3, in the embryo, mos dup4p16.3-15.1/del4p16.3 (placenta P1 eP3) and mos del4p16.2/del4p16.3 (placenta P2) in the placenta were observed, due to non-allelic homologous recombination (NAHR).

**Table 1 genes-14-00668-t001:** NIPT ‘raw data’ indicating a 44.1 duplication of chromosome 4: dup(4)(p16.3p12), in grey.

Variable	Value
fetal_fraction	0.108383051
region_classification	DETECTED: dup(4)(p16.3p12)
chromosome	chr4
start_base	2000001
end_cytoband	p12
region_size_mb	44.1
region_llr_trisomy	148.3858609
region_llr_monosomy	NA
region_t_stat_long_reads	16.18982321
region_mosaic_ratio	0.792664396
region_mosaic_llr_trisomy	148.8731223
region_mosaic_llr_monosomy	NA

## Data Availability

The data that support the findings of this study are available upon reasonable request from the corresponding author. The data are not publicly available due to privacy or ethical restrictions.

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
