# Peer review of "A Case Report of a Feto-Placental Mosaicism Involving a Segmental Aneuploidy: A Challenge for Genome Wide Screening by Non-Invasive Prenatal Testing of Cell-Free DNA in Maternal Plasma"

_genes, 2023, doi:10.3390/genes14030668_

Round 1

Reviewer 1 Report

In this article, the authors describe a case of a 32 year-old pregnant woman with a 44.1 Mb duplication on the short arm of chr.4. Amniocentesis was carried out at 18 weeks of gestation and revelead a de novo deletion of 1.2Mb at chr.4, near to the critical region of Wolf-Hitschhorn syndrome. A normal 46 XY karyotype was identified, but further investigations found discordant results which lead to a true fetal mosaicism of type VI confirmation. Two placental samples were identified with 1.2 Mb deletion + 26.5 Mb duplication in chr.4 in 60% of the cells, and one placental sample, with 1.2 Mb deletion + 4.6 Mb deletion in Chr.4, in 70% of the cells. 

In my opinion, the presented subject is of wide interest, being described so far quite rarely in the specialized literature. The authors clearly, concisely, but comprehensively presented all the elements of anamnesis and diagnosis that they had at their disposal. The case report was accompanied by a review of the scientific literature, a fact that gives a certain value to the presented article.

Only a few minor revisions related to formatting: on row 49 there is an extra space between `biological' and `causes', as well as before `chromosomal' on row 60. On row124, the name of the bibliographic reference must be changed to its corresponding number. On row 197, the round bracket must be deleted.

Otherwise, I congratulate the authors for the work undertaken.

Author Response

Only a few minor revisions related to formatting: on row 49 there is an extra space between `biological' and `causes', as well as before `chromosomal' on row 60.

Response: Thanks for your comments. As you suggested we removed a space between `biological' and `causes' on row 49 and before `chromosomal' on row 60

On row124, the name of the bibliographic reference must be changed to its corresponding number. On row 197, the round bracket must be deleted.

Response: As you suggested we changed the name of the bibliographic reference to its corresponding number on row 124 and we deleted the round bracket on row 197

Reviewer 2 Report

I enclose my comments

Author Response

-Line 45-46:  In relation with clinical impact of additional findings. Several studies demonstrated that the majority of additional finding that are identified through genome wide NIPT have a clinical impact [12].

The authors cited [12] described: Additional findings were detected in 402/110,739 pregnancies (0.36%). For 358 cases, the origin was proven to be either fetal (n = 79; 22.1%), (assumed) confined placental mosaicism (CPM) (n=189; 52.8%), or maternal (n= 90; 25.1%). For the remaining 44 (10.9%), the origin of the aberration could not be determined

My comment is 25.1 % are maternal so exits a clinical impact in but 52.8% of additional findings are due to CMP or fetal.

These number could be confusing because the “majority” could lead to think a big % and most of a CNV are CPM (data shown in Fig. 1).

Response: Thanks for your comments. As the reviewer suggested, we replaced “majority” with “several”

-Line 98: 32 years old is considered in your country an indication for prenatal diagnosis. She is Young?

Response: Thanks for your comments. No, 32 years old is not considered an indication for prenatal diagnosis, so we removed the sentence.

-Lines 131-132: de novo deletion 1.2 Mb. The SNPs array were performed in fetal sample and parents?

Response: Thanks for your comment. We performed SNP-Array analysis in fetal sample and parents, so we added a sentence “as well as on cells from the pheripheral blood of parents” at lane 131-132

-Lines 138-138: Parental karyotypes were normal, but the resolution of karyotype doesn’t allow to detect a deletion of 1.2 Mb

Response: Thanks for your comment. Although the identified deletion  was of 1.2Mb, hence below the resolution of karyotype, it could result from a malsegregation of a structural balanced parental rearrangement involving the chromosome 4.

-Lines 206-209: I think that when a CNV is detected by NIPT is better to study in amniotic fluid and not CVS because the DNA studied by NIPT is the DNA from chronion villi

Response: Thanks for your comment. We agree that when a CNV is detected by NIPT is better to study amniocytes and not CVS. In fact, amniocentesis is considered the gold standard for confirmation of a fetal anomaly, as a CVS procedure analyzes placental tissues only, and cannot be used to rule out a fetal aberration. We discussed this issue at Lane 231-237.

-Fetal autopsy was performed? No congenital anomalies were detected?

Response: Thanks for your comment. We asked for this request, but unfortunately, we received no information about this. We added a sentence in the paper at  lane 161.

Reviewer 3 Report

Authors report a case of mosaioc deletion and duplication in 4p

Report is more or less ok, even though more critical reviews on NIPT should also be considered e.g. PMID: 36428876 or 35986330 in introduction and discussion

Besides authors need to include in their discussion how the reported mosaic may have been formed - a schematic figure to explain that mosaic is necessary. Literature of comparable cases (they are rare but there are reprots of strange mosaics  - not for chr. 4, but for other chromosomes) need to be considered.

Author Response

-Report is more or less ok, even though more critical reviews on NIPT should also be considered e.g. PMID: 36428876 or 35986330 in introduction and discussion

Response: Thanks for your comment. As the reviewer suggested, we added the reference with PMID 35986330 in the Introduction section.

-Besides authors need to include in their discussion how the reported mosaic may have been formed - a schematic figure to explain that mosaic is necessary.

Response: Thanks for your comments. As the reviewer suggested, we explained in the Discussion section how the reported mosaic may have been formed, and also added a schematic figure (Figure 5) in the Results section to show the two hypotheses we discussed.

Lane 252-262: “The latter evidence could suggest the hypothesis of an initial formation of a zygote with del 4p16.3 deletion, originating during the parental gametogenesis; later, in the stages of embryonic development, the other genomic rearrangements observed in samples 1, 2 and 3 of the placenta (del4p16.2 and dup4p16.3), due to non-allelic homologous recombination (NAHR) and, subsequent to a mitotic non-disjunction, would have originated and confined in the placenta (Figure 5A). An alternative hypothesis could be an initial formation of a zygote 46,XY, originating during parental meiotic events. Then, during the mitotic events of embrionic development, the del4p16.3, in the embryo, mos dup4p16.3-15.1/del4p16.3 (placenta P1 eP3) and mos del4p16.2/del4p16.3 (placenta P2) in placenta were observed, due to non-allelic homologous recombination (NAHR) (Figure 5B).

-Literature of comparable cases (they are rare but there are reports of strange mosaics - not for chr. 4, but for other chromosomes) need to be considered.

Response: As the reviewer suggested, we searched for comparable cases in literature, but we did not find them. We added a sentence in Discussion section, lane 267-268: “Moreover we cannot exclude that there may be other mechanisms, since we have not found similar cases in the literature”.

Round 2

Reviewer 3 Report

Thanks for working on the raised points

paper is ok now from my side